# Antimicrobial and Nutritional Potentials of *Bacillus* Strains

**DOI:** 10.3390/ijms26199363

**Published:** 2025-09-25

**Authors:** Yingying Tang, Xiaohua Yu, Yanbing Guo, Ruichao Yue, Jianmin Yuan

**Affiliations:** 1State Key Laboratory of Animal Nutrition and Feeding, College of Animal Science and Technology, China Agricultural University, Beijing 100193, China; sy20243041032@cau.edu.cn (Y.T.); yuxiaohua@tnong.com (X.Y.); 2College of Resources and Environmental Sciences, China Agricultural University, Beijing 100193, China; guoyb@cau.edu.cn; 3National Key Laboratory of Veterinary Public Health and Safety, College of Veterinary Medicine, China Agricultural University, Beijing 100193, China

**Keywords:** *Bacillus*, *Clostridium perfringensinal*, microbiota, growth performance, probiotics

## Abstract

*Bacillus* species represent promising alternatives to antimicrobial growth promoters, offering potential benefits for productivity and gut health in broilers. This study aimed to isolate *Bacillus* strains with inhibitory activity against *Clostridium perfringens* and evaluate their probiotic potential through in vitro and in vivo approaches. In Experiment 1 (in vitro), five strains—*B. siamensis* C66, *B. tequilensis* Y7, *B. velezensis* L15, *B. amyloliquefaciens* C271, and *B. siamensis* C377—were isolated and assessed for stress tolerance, digestive enzyme production, and antimicrobial activity. All strains demonstrated high survival rates under acid and bile stress, produced multiple digestive enzymes, and significantly inhibited the growth of *C. perfringens*. In Experiment 2 (in vivo), 630 day-old male Arbor Acres broilers were randomly assigned to one of seven dietary treatments for 42 days: a negative control (CON, basal diet), a positive control (ANfT, basal diet supplemented with 6.4 g/t virginiamycin), and five groups receiving basal diet supplemented with one of the *Bacillus* strains at 1 × 10^11^ CFU/kg. Among these, *B. amyloliquefaciens* C271 significantly increased breast muscle yield (*p* < 0.05), improved jejunal morphology—evidenced by increased villus height and villus height-to-crypt depth ratio (*p* < 0.05)—and positively modulated cecal microbiota composition compared to the CON group. These findings demonstrate that the newly isolated *B. amyloliquefaciens* C271 possesses strong probiotic properties in vitro and promotes growth performance and gut health in broilers, suggesting its viability as an antibiotic growth promoter substitute.

## 1. Introduction

*Clostridium perfringens* is an anaerobic zoonotic pathogen that is widely distributed in natural environments and the digestive tracts of animals [1]. *C. perfringens* is the primary agent responsible for necrotic enteritis (NE) in poultry, leading to intestinal bleeding, mucosal necrosis, growth retardation, and even high mortality rates, resulting in significant economic losses [2]. Historically, antibiotics were frequently employed as efficacious substances incorporated into feed for the treatment of NE. In recent years, the overuse of antibiotics has led to increased antibiotic resistance in *C. perfringens*, posing a serious threat to the promotion of antibiotic-free farming [3]. This limitation has spurred the investigation of various antibiotic alternatives for preventing *C. perfringens* infections, such as acidifiers [4], polysaccharides [5], probiotics [6], herbal extracts [7], and fatty acids [8].

Among these alternatives, probiotics have demonstrated considerable benefits for improving growth, immune function, and gut health of livestock and poultry [9,10,11]. Probiotics currently used in animal production primarily include *Bacillus*, *Bifidobacterium*, and *Lactobacillus*, with *Bacillus* considered the most promising owing to its widespread availability, strong stress tolerance, and capacity to produce various digestive enzymes and antibacterial compounds [12,13]. Predominant *Bacillus* spp. studied as monogastric probiotics include *B. subtilis*, *B. licheniformis*, *B. coagulans*, *B. amyloliquefaciens*, *B. velezensis*, and *B. cereus*, typically isolated from soil or animal gastrointestinal tracts [14]. Supplementation with *Bacillus* probiotics has been shown to enhance the growth performance, meat quality, antioxidant capacity, and immune function of broiler chickens, as well as reduce intestinal damage and inflammatory responses triggered by *C. perfringens* infection [15,16,17,18]. Although numerous *Bacillus* probiotics have been reported, there are few studies on specific *Bacillus* strains that can effectively inhibit pathogenic bacteria, and the sources of these strains are limited. There remains a need to identify novel *Bacillus* strains with enhanced antibacterial efficacy and broader functional features for use in modern livestock production.

Therefore, this study aimed to isolate and evaluate novel *Bacillus* strains with potent activity against *C. perfringens*. In this study, we isolated *Bacillus* strains from the intestinal contents of diverse animal species by leveraging their heat-resistant and spore-forming properties. Through in vitro screening, five *Bacillus* strains demonstrating effective inhibition of *C. perfringens* growth were selected. We then evaluated the probiotic properties of these antimicrobial *Bacillus* strains, including acid tolerance, bile salt resistance, antibiotic susceptibility, and enzyme production capabilities. Broiler trials were conducted to evaluate the impact of these five strains on growth performance and gut health, and to further explore their in vivo probiotic functions.

## 2. Results

### 2.1. Isolation, Screening and Identification of Bacteriostatic Bacillus Strains

The results of the strains isolated from chicken, duck, deer and gadfly are shown in Table 1. A total of 1437 bacterial strains demonstrating in vitro inhibitory activity against *C. perfringens* were isolated using the double-layer agar primary sieving. Primary screening identified 657 *Bacillus* spp. isolates, and secondary screening selected 36 strains demonstrating strong inhibitory activity.

After 16S rRNA identification, five strains of *Bacillus* spp. were obtained: *B. siamensis* C66, *B. amyloliquefaciens* C271, and *B. siamensis* C377 from black gadfly (Gadfly niger) intestines; *B. tequilensis* Y7 from duck cecum; and *B. velezensis* L15 from deer feces (Table 2). Their 16S rRNA sequences have been submitted to the NCBI database, with accession numbers PX285217, PX285219, PX285221, PX285218, and PX285220, respectively.

### 2.2. Results of Bacteriostatic Test of the Bacillus Strains

The results of the bacteriostatic test of 5 strains of *Bacillus* are shown in Table 3. Using LB broth without antibiotics as the negative control, the fermentation supernatants of different *Bacillus* strains were uniformly spread at the same concentration on agar plates containing *C. perfringens.* After overnight incubation, the diameters of the inhibition circle were more than 15 mm, which indicated that the five *Bacillus* species demonstrated potent in vitro growth suppression against *C. perfringens*.

### 2.3. Results of Enzyme Production of the Screened Bacillus Strains

The results of the enzyme production test of 5 strains of *Bacillus* are shown in Table 4. *B*. *siamensis* C66, *B*. *tequila* Y7, *B*. *amyloliquefaciens* C271, *B*. *siamensis* C377, and *B*. *beleriensis* L15 can produce amylase and protease. The diameter of the transparent circle formed by the five strains of *Bacillus* amylase and protease was more than 15 mm, which indicated that they had a good ability to produce amylase and protease.

### 2.4. Tolerance Test of the Screened Bacillus Strains

After treatment with hydrochloric acid and bile salts, the survival rates of all five strains exceeded 90% (Table 5), indicating that all five strains of *Bacillus* exhibit good tolerance to hydrochloric acid and bile salts.

### 2.5. The Effects of Adding the Screened Bacillus Strains to Diets on the Growth Performance and Breast Muscle Indexes of Broilers

Table 6 shows that in the late feeding stage (days 22–42), the supplementation of C66, Y7, and C377 significantly increased ADFI and FCR (*p* < 0.05) versus CON, whereas C271 and L15 had no statistically significant effect on ADFI but significantly improved the FCR (*p* < 0.05). During the 42-day trial, the FCR was significantly enhanced in the groups supplemented with antibiotics and *Bacillus* strains (*p* < 0.05) versus CON. Notably, C377 supplementation surpassed antibiotic efficacy, whereas the remaining four *Bacillus* groups performed comparably to the antibiotic-treated.

Table 7 reveals that dietary supplementation with C271 significantly increased the 42-day breast muscle index of broilers relative to CON (*p* < 0.05). Supplementation with C66, Y7, and L15 showed effects consistent with the ANT group, with no statistically significant impact on the breast muscle index.

### 2.6. The Effects of Adding the Screened Bacillus Strains to Diets on the Immune Performance

Based on comprehensive growth performance and slaughter performance, groups C271 and C377 were selected for subsequent data analysis. As presented in Table 8, the bursa index at 42 days of age was significantly reduced (*p* < 0.05) in broilers supplemented with antibiotics or C271 compared with the CON group. C377 supplementation did not statistically significantly alter the bursal index in 42-day-old broilers. Furthermore, the cecal index was significantly lowered (*p* < 0.05) by dietary supplementation with C377.

As shown in Table 9, antibiotics and other *Bacillus* strains had no statistically significant effect on serum IgA or IgM levels at either 21 or 42 days of age. Notably, C377 supplementation significantly reduced serum IgY levels at 42 days (*p* < 0.05).

### 2.7. Effects of Different Bacillus Strains Additions to the Diet on the Morphological Structure of the Jejunum in the Broilers

In 42-day-old broilers (Table 10), C271 supplementation significantly enhanced both jejunal VH and VH/CD ratio relative to both CON and ANT (*p* < 0.05). Notably, C377 supplementation significantly reduced jejunal crypt depth while improving the VH/CD ratio versus ANT (*p* < 0.05).

### 2.8. Effects of Adding Different Bacillus Strains to the Diet on the Expression of Inflammatory Factors in the Jejunal Mucosa of Broilers

The influence of different *Bacillus* additives on the expression of jejunal mucosal inflammatory factors in 42-day-old broilers is presented in Table 11. Neither *Bacillus* supplementation nor antibiotic treatment statistically significantly modulated IL-2, IL-10, IL-8, IL-6, IFN-γ, or TNF-α levels.

### 2.9. The Effect of Different Bacillus Strains on the Microbial Flora of the Cecum of Broilers

Species number, Shannon, Simpson, Chao1, and Abundance-based Coverage Estimator (ACE) are important indicators used to assess the α diversity of cecal microbial diversity. While *Bacillus* supplementation showed no statistically significant impact on Shannon or Simpson indices, C271 and C377 treatments significantly lowered species richness, Chao1, and ACE values (*p* < 0.05), mirroring the ANT group (Table 12).

PCoA based on unweighted UniFrac distances was employed to visualize differences in cecal microbiota β-diversity among treatment groups (Figure 1). The resulting ordination revealed a clear separation between two clusters: one comprising the CON and C271 groups, and another including the ANT and C377 groups. Statistical validation using pairwise PERMANOVA with FDR correction supported these observations (Table 13). Specifically, no significant difference was detected between the CON and C271 groups (q = 0.278), indicating similar microbial community structures. The ANT group showed significant divergence from the CON group (q = 0.003), highlighting the substantial impact of antibiotic treatment. Notably, although the ANT and C377 groups exhibited similar microbiota structures and clustered together visually, this apparent similarity did not reach statistical significance (q = 0.083).

Based on these findings, further analysis was conducted on bacterial relative abundance at different taxonomic levels. The cecal microbiome in 42-day-old broilers was dominated by three primary phyla: Firmicutes, Bacteroidetes, and Proteobacteria (Figure 2A). Supplementation with antibiotics, C271, and C377 increased the relative abundance of Bacteroidetes and Verrucomicrobiota. Figure 2B revealed that *Barnesiella*, *Bacteroides*, and *Enterobacteriaceae* represented the predominant genera. Dietary supplementation with five distinct *Bacillus* strains was found to increase the relative abundance of anaerobic bacteria, including *Bacteroidaceae*, *Ruminococcaceae*, and *Lactobacillus* in the cecal microbiota (Figure 2C), producing effects comparable to those observed in the antibiotic group.

LEfSe analysis of ASV data (LDA score > 4) was utilized as an exploratory approach to identify potential taxonomic biomarkers across treatment groups (Figure 2D). After applying false discovery rate (FDR) correction for multiple comparisons (Table 14), several taxa showed statistically significant enrichment patterns: the CON group was significantly enriched for Gammaproteobacteria (LDA = 4.15, q = 0.012); the ANT group showed significant enrichment of the kingdom Bacteria (LDA = 4.36, q = 0.009) and the genus Romboutsia (LDA = 4.07, q = 0.018); the C271 group was significantly enriched for the kingdom Bacteria (LDA = 4.23, q = 0.014); and the C377 group showed significant enrichment of the genus Alistipes (LDA = 4.10, q = 0.010).

## 3. Discussion

Enterocolitis induced by *C. perfringens* in poultry has become a major public health concern [19]. In light of antibiotic overuse and escalating resistance, the development of effective agents against *C. perfringens* was imperative for the treatment of associated diseases in the future. *Bacillus* is a well-known beneficial bacterium that is extensively used in animal feed additives due to its broad-spectrum antibacterial properties and relatively high activity [20,21]. The present study screened five strains of *Bacillus* exhibiting strong *C. perfringens* inhibition and proceeded to analyze their probiotic properties. These five strains were then incorporated into broiler diets to investigate their impact on the chickens’ growth performance and gut health.

*Bacillus* bacteria can form spores to withstand adverse environmental conditions, enabling survival in high temperatures, acidic, alkaline, and saline environments [22]. This study took advantage of their characteristics by using high-temperature screening, which greatly improved screening efficiency. The Oxford Cup experiment was typically employed to assess the antimicrobial activity of various materials against pathogens [23]. In vitro antibacterial testing revealed that five *Bacillus* strains exhibit excellent antibacterial activity against *C. perfringens*, with antibacterial zone diameters exceeding 15 mm. The antimicrobial activity of *Bacillus* spp. may derive from bioactive metabolites secreted by viable bacteria. Previous studies have demonstrated that *Bacillus* culture supernatants exert potent inhibitory effects against *C. perfringens* by disrupting cell wall integrity, inducing cytoplasmic condensation and vacuolization [16,24].

Probiotics added to feed must pass through the gastrointestinal tract and proliferate within it. The acidic and alkaline environments in the gastrointestinal tract generally inhibit bacterial growth and activity, thereby affecting the efficacy of probiotics [25]. After 2 h of treatment with hydrochloric acid and bile salts, the survival rate of the five *Bacillus* strains remained above 90%, indicating good tolerance to the gastrointestinal environment. Additionally, the ability to produce enzymes is also one of the criteria for evaluating probiotics. Proteases and amylases are enzymes that break down proteins and carbohydrates into amino acids, aldehydes and amines [26]. All five *Bacillus* strains in this experiment can produce amylase and protease. Although broilers naturally produce digestive enzymes, the production of additional enzymes by gut probiotics further aids carbohydrate and protein digestion, thereby enhancing nutrient absorption [27].

Previous research confirms that dietary supplementation with *Bacillus* strains as probiotics in broiler feed improves feed utilization efficiency and enhances growth performance [28,29]. However, this study found that, relative to CON, both the ANT group and the five *Bacillus* groups significantly improved the FCR of 42-day-old broilers, but did not influence their average body weight and BWG. The addition of *Bacillus* or antibiotics may induce changes in the cecal microorganisms, causing amino acids to be prioritized for microbial protein synthesis rather than muscle deposition [30]. Furthermore, studies have shown that dietary supplementation with *B. amyloliquefaciens* did not significantly improve growth performance at 9 weeks of age, but enhanced growth parameters in red-feathered native chickens by 11 weeks of age [31]. The 1–42 day observation period selected in this study may not have captured the stage-specific characteristics of body weight gain in broilers. The breast muscle index serves as a critical measure of broilers’ growth performance. Dietary supplementation with C271 significantly increased breast muscle yield and improved slaughter performance in broilers. Meanwhile, dietary inclusion of C66, Y7, and L15 showed a tendency to enhance 42-day breast muscle yield, demonstrating comparable effects to antibiotic treatment. These findings corroborate prior studies showing that dietary supplementation with *Bacillus subtilis*–enzyme complex increased breast muscle percentage in broilers [32].

The thymus, spleen, and bursa of Fabricius constitute pivotal immune structures in broilers, and their immune organ indices serve as reliable indicators of immunological responsiveness in broilers. Notably, the bursa exhibits greater sensitivity to dietary interventions compared to other organs in broilers [33]. This study demonstrated that ANT and C271 treatments significantly reduced the bursa of Fabricius index in 42-day-old broilers, while C377 had no significant effect. Zou et al. [34] reported that broilers fed multi-strain probiotics showed significantly increased spleen index, while thymus and bursa of Fabricius indices remained unchanged. Other studies have demonstrated that *Bacillus* supplementation had no effect on thymus and spleen weights but increased the bursa of Fabricius index [32]. These differential effects may result from variations in bacterial strains, dosages, and host status, which can lead to either immunostimulatory or immunosuppressive outcomes.

Serum immunoglobulin reflects the humoral immune status of broilers. The main mechanisms of probiotics’ immune-modulating effects include: increasing the production of antimicrobial peptides [35], neutralizing dysbiosis [36], stimulating mucosal immunity [37], and increasing the production of specific antibodies [38]. The study found that supplementation with C271 tended to increase serum IgA concentration in 21-day-old broilers. This finding aligns with a previous study demonstrating that supplementation with *B. subtilis* enhanced IgA and IgY levels in chickens [39]. Notably, dietary supplementation with C377 significantly reduced serum IgY levels in 42-day-old broilers. The decreased IgY concentration might be attributed to the probiotic’s elimination of pathogens that would otherwise stimulate natural antibody production [40]. The immune system-derived inflammatory cytokines serve crucial functions in protecting against bacterial and viral infections while balancing immunological homeostasis [41]. The current results revealed that dietary *Bacillus* supplementation had no significant effect on inflammatory cytokine expression in 42-day-old broilers, consistent with the observed reduction in the bursa of Fabricius index. Due to the addition of exogenous *Bacillus* and antibiotics, the microbial diversity in the intestines of broilers decreases, and the immune system’s mobilization requirements are reduced.

Broiler growth rate is strongly correlated with intestinal morphogenesis, where increased VH and VH/CD ratio reflect greater absorptive surface area and improved nutrient uptake efficiency [42]. Existing research indicates that the dietary addition of *B. subtilis* significantly enhanced jejunal VH and VH/CD ratio, concurrently decreasing CD in 21-day-old broilers [43]. In this study, C271 supplementation significantly improved jejunal VH and VH/CD ratio in broilers at 42 days of age (*p* < 0.05). Meanwhile, C377 reduced jejunal CD and increased VH/CD ratio in 42-day-old broilers (*p* < 0.05). *Bacillus* strains could enhance jejunal structure in broilers through intestinal tissue repair and beneficial microbiota colonization [44].

Gut microorganisms perform critical host functions: digesting nutrients, preventing pathogen colonization, strengthening intestinal barrier properties, and stimulating immune system development [45]. Existing research indicates that *B. licheniformis* fermentation products reduced fecal bacterial diversity in broilers [46], while *B. subtilis*-fermented feed decreased cecal microbial richness [47]. These findings are partially consistent with our results, where supplementation with C271 and C377 reduced bacterial richness and diversity in the cecum. This may be related to the antibacterial activity of the added *Bacillus* strains. PCoA demonstrated two primary clusters: one comprising control and C271 with conserved microbiota, and another containing antibiotic and C377 groups, indicating C377’s antibiotic-mimicking effects on microbial composition, possibly due to antimicrobial action.

Firmicutes and Bacteroidetes constituted the predominant phyla in broiler cecal communities. Similar findings were also reported in cecal samples from probiotic-supplemented broilers [48,49]. Supplementation with C271 and C377 increased the relative abundance of Bacteroidetes. Multiple species of this phylum contribute to intestinal nutrient metabolism and are capable of fermenting carbohydrates to produce short-chain fatty acids [50]. It is worth noting that supplementation with C271 and C377 can increase the relative abundance of Verrucomicrobiota. Studies have shown that Verrucomicrobiota can improve diverse intestinal indicators, including VH, VH/CD, and intestinal length [51,52]. Aligning with these results, our study found a significant increase in VH/CD in the jejunum after dietary inclusion with C271 and C377. Dietary supplementation with different *Bacillus* strains increased the relative abundance of anaerobic microbes, including Bacteroides, Alistipes, and Lactobacillus in the cecal microbiome of 42-day-old broilers. Lactobacillus strains are generally considered beneficial for gut and host health due to their immunomodulatory capacity, pathogen inhibition, and bacteriocin synthesis [53]. The increased abundance of Bacteroides and Alistipes suggests enhanced fermentation capacity and improved gut health. The LEfSe analysis, following rigorous multiple-testing correction, revealed distinct strain-specific modulation of cecal microbiota. Although the C271 group shared overall structural similarity with CON, its unique enrichment at the kingdom level, Bacteria (LDA = 4.23, q = 0.014), suggests that *B. amyloliquefaciens* C271 may induce a broad, nonspecific stimulation of bacterial biomass. The specific enrichment of Alistipes in the C377 group (LDA = 4.10, q = 0.010), combined with its clustering tendency with ANT in β-diversity analysis, strongly indicates that *B. siamensis* C377 may shape a microbial environment partially resembling that of antibiotic treatment, likely through its antimicrobial activity. These findings underscore the critical importance of strain-specific selection for achieving targeted modulation of the gut microbiota.

## 4. Materials and Methods

### 4.1. Animal Ethics Statement

The study protocol received ethical approval from the Laboratory Animal Ethical Committee at China Agricultural University (Beijing, China, Approval no. AW71405202-1-02) and conformed to the ARRIVE guidelines.

### 4.2. Isolation and Screening of Bacillus Cereus

Cecal content samples were collected from 20 chickens and 15 ducks, alongside 8 deer fecal samples and 5 intestinal content samples from black soldier fly larvae, serving as sources for the initial isolation of *Bacillus* strains. After heat treatment (85 °C for 10 min), the samples were homogenized in sterile saline, shaken for 5 min, and serially diluted. Diluted aliquots were cultured on LB agar (Hopebio Biotechnology Co., Ltd., Qingdao, China) at 37 °C for 16 h. Colonies were selected by morphology and Gram staining, with purified cultures cryopreserved at −80 °C for further analysis. *Bacillus* spp. with inhibitory activity against *C. perfringens* were initially screened using the bilayer culture method on Reinforced Clostridial Medium (RCM; Hopebio Biotechnology Co., Ltd., Qingdao, China), followed by secondary selection via the Oxford cup assay to identify superior inhibitory strains. Each group of experiments was repeated three times. Strains showing clear inhibition zones were selected for further testing.

### 4.3. Identification of Strains

Genomic DNA from antibacterial strains was extracted with a commercial bacterial DNA extraction kit (Tengen Biotechnology, Beijing, China). The 16S rRNA region was PCR-amplified with universal primers 27F: 5′-AGAGTTTGATCCTGGCTCAG-3′ and 1492R: 5′-GGTTACCTTGTTACGACTT-3′, and subsequently sequenced by Shanghai Sangong Bioengineering Co. (Shanghai, China). Comparative analyses were performed using the BLASTn tool in the NCBI database (https://blast.ncbi.nlm.nih.gov, accessed on 1 July 2025) to determine the affinities between the selected *Bacillus* spp. and other strains.

### 4.4. Acid and Bile Resistance Testing

Strain tolerance was assessed following Soni et al. [54] with adjustments. LB medium was acidified to pH 3.0 with 1 M HCl (pH 7.1 as control). Five *Bacillus* strains (1% inoculum) were cultured in both media at 37 °C, 220 rpm for 8 h. Survival rates (%) were quantified via plate counting (100 μL spread plates), with results expressed as survival percentage relative to the control. Each group of experiments was repeated three times.

Bile salt solutions (0.30%) were prepared in sterile LB broth and filter-sterilized. Activated cultures (1% inoculum) were added to LB medium containing bile salt and incubated (37 °C, 220 rpm, 24 h). Bacterial growth was assessed by plate counting and expressed as a percentage relative to bile salt-free controls. Each group of experiments was repeated three times.

### 4.5. Enzyme Production Testing

To evaluate extracellular enzyme production, we referred to the method of Zhang et al. [55] with adjustments. 2 μL aliquots of 16 h *Bacillus* cultures were spot-inoculated onto soluble starch agar and casein agar. Following overnight inverted incubation at 37 °C, plates were examined for hydrolytic clearance zones. Each group of experiments was repeated three times.

### 4.6. Animal Testing

#### 4.6.1. Experimental Design and Animal Management

In this study, 630 day-old AA male broilers were randomly allocated to 7 dietary treatments, with each replicate within a group having a similar body weight (Table 15). Each group had 5 replicates, and each replicate had 18 chickens, which were reared flat on the ground with thick bedding. The isolated *Bacillus* strains were activated and inoculated into LB broth medium, and then cultured at 37 °C and 180 rpm for 16 h. Subsequently, the starter culture was transferred at a 5% (*v*/*v*) inoculum rate to a larger volume of LB medium for 24 h of fermentation. The fermented broth was centrifuged to collect a concentrated bacterial pellet, which was then freeze-dried using a freeze dryer until a stable dry powder was obtained. The freeze-dried powder was thoroughly mixed into the base feed to achieve a final concentration of 1 × 10^11^ CFU per kilogram of feed.

#### 4.6.2. Diet Formulation, Housing, and Management

In this study, a corn–soybean meal–wheat-type basal diet was provided in two phases: starter (1–21 d, mash) and grower (22–42 d, pellet). Table 16 presents the nutritional levels and ingredient composition of the basal diets. The experiment was conducted at the environmentally controlled poultry research facility of the China Agricultural University (Zhuozhou Base) and lasted for 42 d.

The birds were raised in floor pens bedded with fresh wood shavings, with ad libitum access to feed and water. Feed was provided using hanging tube feeders (Model XYZ, ABC Husbandry Equipment Co., Ltd., Beijing, China), and water was supplied via a nipple drinking system (Model NDP-2000, DEF Poultry Systems, Guangzhou, China). The room temperature was maintained at 33 °C during the first 4 days and was gradually decreased by approximately 2.5 °C per week until reaching a final temperature of 23 °C. The relative humidity was controlled at 50–65% throughout the trial period. Heating, cooling, and ventilation were automatically regulated by a central environmental control system (Model ECP-8000, GHI Climate Control Inc., Jinan, China) to maintain adequate air quality, with ammonia levels kept below 10 ppm. A standard lighting program was implemented using LED white lights (Model Poultry-LED-20W, JKL Lighting Technology Co., Ltd., Hangzhou, China): 23 h of light at 20–30 lux intensity and 1 h of darkness for the first 8 days, followed by 16 h of light and 8 h of darkness from day 9 until the end of the trial. All birds received routine vaccinations following standard commercial protocols. The house was thoroughly cleaned and disinfected prior to chick arrival using a broad-spectrum disinfectant containing quaternary ammonium compounds (Sanisept Poultry, MNO Biosecurity Solutions, Shanghai, China). Bedding was managed to remain dry and clean throughout the experimental period.

#### 4.6.3. Sample Collection

At 21 and 42 d, one representative bird (nearest to the mean weight) was selected from each replicate pen for sampling, resulting in *n* = 5 birds per treatment group at each time point. Jugular venipuncture was performed, with serum isolated and cryopreserved at −20 °C. Following electrical stunning, euthanasia was performed via exsanguination. The spleen, bursa, and unilateral pectoral muscle were excised and precisely weighed. A jejunal portion (2 cm) was immersion-fixed in 4% paraformaldehyde for intestinal morphological analysis. Jejunal mucosa was scraped, cryopreserved in liquid nitrogen, and stored at −80 °C. Cecal digesta samples were collected and archived at −80 °C for microbial analysis.

#### 4.6.4. Measurement of Growth Performance Indexes

Body weights of broilers at 1, 21, and 42 days were measured on a per-cage basis. Feed intake was monitored daily during the trial. The average daily feed intake (ADFI), feed conversion ratio (FCR), body weight gain (BWG) and body weight were calculated based on the method described by Huang et al. [56].

#### 4.6.5. Measurement of Organ Index and Slaughtering Performance

The spleen, bursa and appendix were carefully excised and precisely weighed. The organ index was calculated as follows: Immune organ index = immune organ weight (g)/broiler body weight (kg) [56].

At 42 days of age, the breast muscle of one side of the broiler was accurately weighed. The breast muscle index was calculated as follows: Breast muscle index = breast muscle weight (kg)/live weight (kg) [56].

#### 4.6.6. Morphological Structure of the Intestine

Jejunal samples of broilers at 21 and 42 d were taken for fixation (4% paraformaldehyde). The fixed jejunal samples were embedded, stained, and sectioned according to the method of Luo et al. [57] for histological evaluation of intestinal morphology and structure. 10 villi were selected from each slice, and the villus height (VH) and crypt depth (CD) were measured to calculate the ratio of villus height/crypt depth (VH/CD), and to compare the development of jejunal villus morphology and structure of the broilers in each group.

#### 4.6.7. mRNA Expression of Intestinal Inflammatory Factors

Jejunal mucosal samples were collected from 21- and 42-day-old broilers, with total RNA extracted via Trizol reagent. The quality of RNA extraction was verified with a Nano-300 microspectrophotometer (Beijing YuanPingHao BioTech Co., Ltd., Beijing, China), and cDNA was reverse transcribed using a Primer Script™RT kit (Perfect Real Time; Takara Biotechnology Co., Ltd., Beijing, China). RT-PCR was then performed on an Applied Biosystems 7500 Fast Real-Time PCR system using the SYBR Premix Ex Taq™ kit (Takara Biotechnology Co., Ltd., Beijing, China). The relative expression of IL-2, IL-6, IL-8, IL-10, IFN-α, IFN-γ and IL-1β was measured with β-actin as the reference gene. Primer sequences are listed in Table 17. All primers used in this study were synthesized by Beijing Qingke Biotechnology Co., Ltd. (Beijing, China).

#### 4.6.8. Serum Immunity Factors

Serum concentrations of IgA, IgY and IgM were quantified with the commercial ELISA kits (Solarbio Science & Technology Co., Ltd., Beijing, China). The kit catalog numbers were as follows: IgY (SEKH-0016), IgA (SEKH-0017), and IgM (SEKH-0018).

#### 4.6.9. 16S rRNA Sequencing of Cecum Microorganisms

A total of 35 cecal content samples were collected at the end of the trial (42 days of age) specifically for microbial analysis. Microbial community DNA was isolated using the E.Z.N.A.^®^ Soil DNA Kit (Omega Bio-Tek Inc., Norcross, GA, USA) per manufacturer’s guidelines. The quality and concentration of extracted DNA were verified by 1% agarose gel electrophoresis. Amplification of bacterial 16S rRNA V3-V4 hypervariable domains by PCR, using primers 338F (5′-ACTCCTACGGGAGGCAGCAG-3′) and 806R (5′-GGACTACHVGGGTWTCTAAT-3′). The DNA sequencing was performed on the Illumina NovaSeq 6000 platform by Novogene Co., Ltd. (Beijing, China). Operational Taxonomic Units (OTUs) were delineated at 97% similarity with Uparse (v7.0.1001) and annotated against the SILVA SSU v138 reference database [58,59]. Amplicon Sequence Variants (ASVs) were inferred using the DADA2 algorithm (v1.12.1). Alpha diversity of the cecal microbiota was evaluated by calculating the Observed species number, Chao1, ACE, Shannon, and Simpson indices [60]. Beta diversity was assessed using principal coordinate analysis (PCoA) based on the Unweighted UniFrac distance metric [61]. To statistically evaluate group differences in microbial community structure, pairwise PERMANOVA tests with 999 permutations were performed on the Unweighted UniFrac distance matrix [62]. The resulting *p*-values were adjusted for multiple comparisons using the Benjamini–Hochberg false discovery rate (FDR) method. Both α-diversity (within-sample) and β-diversity (between-sample) indices were calculated according to the ASV table. The rarefaction depth was set to 42,000 sequences per sample. These analyses were performed using the QIIME 2 pipeline (v2023.2) [63]. Potential differentially abundant taxa across groups were preliminarily identified using Linear Discriminant Analysis Effect Size (LEfSe) [64]. The analysis was performed on relative abundance-transformed data using the standard LEfSe workflow on the Majorbio Cloud Platform (https://cloud.majorbio.com, accessed on 2 July 2025), which applies internal clipping normalization and unadjusted pairwise Wilcoxon tests following a significant Kruskal–Wallis result. To address the concern of multiple comparisons, we have now applied Benjamini–Hochberg FDR correction to the LEfSe results [65].

### 4.7. Data Analysis

Statistical analysis was performed using one-way ANOVA in SPSS 22.0 software [66]. Prior to ANOVA, all datasets were tested for normality using the Shapiro–Wilk test and for homogeneity of variances using Levene’s test; both assumptions were satisfied. Upon obtaining a significant main effect, post hoc comparisons were conducted using Tukey’s Honestly Significant Difference (HSD) test to control the family-wise Type I error rate across all pairwise comparisons. Quantitative data are presented as mean ± standard error of the mean (SEM) (Table 3, Table 4 and Table 5) or as mean values (Table 6, Table 7, Table 8, Table 9, Table 10, Table 11 and Table 12). Differences were considered statistically significant at *p* < 0.05 and q < 0.05.

## 5. Conclusions

In conclusion, this study identified five *Bacillus* strains—*B. siamensis* C66, *B. tequilensis* Y7, *B. amyloliquefaciens* C271, *B. velezensis* L15, and *B. siamensis* C377—that exhibited potent inhibitory activity against *C. perfringens* in vitro. Comprehensive characterization demonstrated their strong capabilities in amylase and protease production, along with high stress tolerance. In subsequent broiler feeding trials, *B. amyloliquefaciens* C271 emerged as the most effective strain, significantly enhancing breast muscle yield, enhancing VH and VH/CD ratio, optimizing intestinal morphology, and modulating gut microbiota composition.

## Figures and Tables

**Figure 1 ijms-26-09363-f001:**
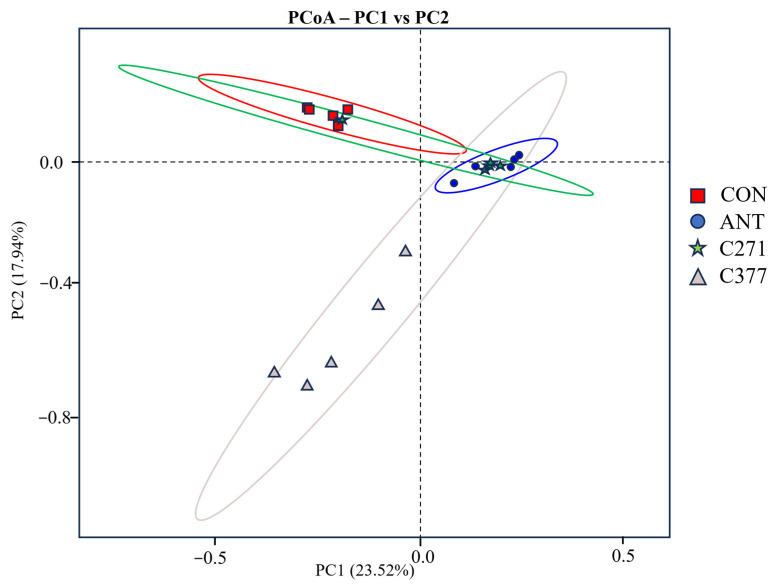
β diversity of cecal microflora. Coils of different colors represent different groups of microbial communities: CON (red), ANT (blue), C271 (green), and C377 (grey).

**Figure 2 ijms-26-09363-f002:**
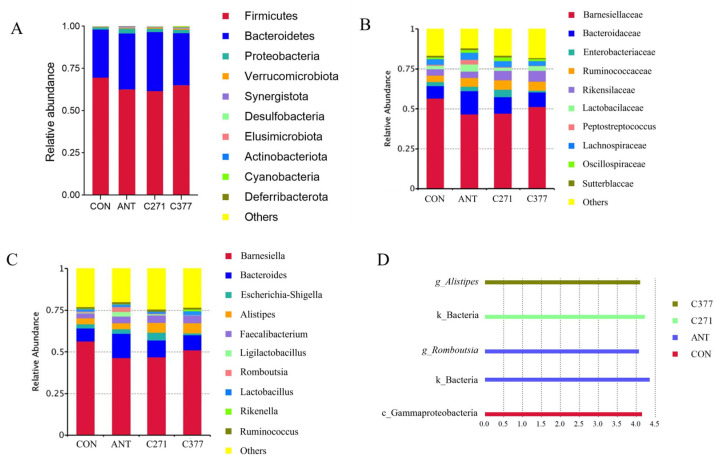
The effect of different *Bacillus* strains on microbial structure and composition of broilers. (**A**) Relative abundance of cecal microorganisms at the phylum level. (**B**) Relative abundance of cecal microorganisms at the family level. (**C**) Relative abundance of cecal microorganisms at the genus level. (**D**) LEfSe analysis diagram of cecal microorganisms.

**Table 1 ijms-26-09363-t001:** Preliminary screening results of *Bacillus* isolates.

Source	Swine Manure	Poultry Cecal Content	Duck Cecal Content	Deer Manure	Total
Initial isolates	671	346	161	259	1437
Primary screened	344	92	78	143	657
Secondary screened	21	6	4	5	36

**Table 2 ijms-26-09363-t002:** Identification results of 16S rRNA sequencing for five bacterial strains.

Strain	Identification Result	Similarity
C66	*Bacillus siamensis*	99%
Y7	*Bacillus tequilensis*	99%
C271	*Bacillus amyloliquefaciens*	99%
L15	*Bacillus velezensis*	99%
C377	*Bacillus siamensis*	99%

**Table 3 ijms-26-09363-t003:** Antibacterial activity of sterile fermentation supernatants from five *Bacillus* strains.

Strain	Inhibition Zone Diameter (mm)
C66	15.60 ± 1.20
Y7	16.30 ± 0.80
C271	18.30 ± 0.50
L15	17.30 ± 0.60
C377	18.20 ± 0.80
LB	8.00 ± 0.00

**Table 4 ijms-26-09363-t004:** Enzyme activity halo zone results of five *Bacillus* strains.

Strain	Amylase Halo Diameter (mm)	Protease Halo Diameter (mm)
C66	15.63 ± 0.20	19.72 ± 0.40
Y7	14.52 ± 0.80	20.35 ± 1.20
C271	14.38 ± 0.60	20.35 ± 1.20
L15	13.32 ± 0.30	15.01 ± 1.10
C377	14.16 ± 0.30	21.36 ± 1.30

**Table 5 ijms-26-09363-t005:** Tolerance test results of five *Bacillus* strains.

Strain	Survival Rate in Simulated Gastric Fluid (%)	Survival Rate in Simulated Intestinal Fluid (%)
C66	93.40 ± 3.30	94.20 ± 3.50
Y7	98.40 ± 2.60	92.30 ± 4.30
C271	98.20 ± 3.70	94.40 ± 2.60
L15	99.50 ± 4.50	95.50 ± 3.80
C377	99.60 ± 1.50	94.60 ± 2.50

**Table 6 ijms-26-09363-t006:** Effects of different *Bacillus* strains supplementation in diets on the performance of broilers.

Items	Body Weight/kg		ADFI/kg			FCR			BWG/kg		
	21 d	42 d	0–21 d	22–42 d	0–42 d	0–21 d	22–42 d	0–42 d	0–21 d	22–42 d	0–42 d
CON ^1^	0.74	2.72	1.05	2.95 ^b^	4.08 ^b^	1.50	1.49 ^c^	1.48 ^c^	0.70	1.98	2.68
ANT ^2^	0.78	2.76	1.08	3.42 ^a^	4.46 ^a^	1.46	1.73 ^ab^	1.64 ^b^	0.74	1.98	2.72
C66	0.72	2.65	1.03	3.41 ^a^	4.38 ^ab^	1.51	1.76 ^a^	1.68 ^ab^	0.68	1.93	2.61
Y7	0.75	2.69	1.07	3.38 ^a^	4.39 ^ab^	1.57	1.72 ^ab^	1.65 ^b^	0.68	1.98	2.66
L15	0.73	2.66	1.07	3.22 ^ab^	4.25 ^ab^	1.56	1.66 ^b^	1.62 ^b^	0.69	1.93	2.62
C271	0.73	2.61	1.05	3.26 ^ab^	4.29 ^ab^	1.53	1.74 ^ab^	1.67 ^ab^	0.69	1.88	2.59
C377	0.73	2.59	1.05	3.33 ^a^	4.37 ^ab^	1.52	1.79 ^a^	1.72 ^a^	0.69	1.86	2.55
SEM	0.008	0.025	0.012	0.028	0.047	0.012	0.018	0.014	0.008	0.064	0.025
*p*-value	0.497	0.584	0.962	0.028	0.103	0.251	<0.01	<0.01	0.521	0.978	0.591

^a–c^ Within a column, means without a common superscript differ significantly (*p* < 0.05). ^1^ CON: negative control (basal diet), the same as the table below. ^2^ ANT: positive control (basal diet + 6.4 g/t virginiamycin), the same as the table below.

**Table 7 ijms-26-09363-t007:** Effects of different *Bacillus* strains supplementation on the breast muscle index of broilers at 42 days of age.

Items	Live Weight (kg)	Breast Muscle Weight (kg)	Breast Muscle Index (%)
CON	2.99	0.503	16.85 ^b^
ANT	3.02	0.553	18.31 ^ab^
C66	2.94	0.529	17.99 ^ab^
Y7	2.97	0.653	18.29 ^ab^
L15	2.85	0.518	18.17 ^ab^
C271	2.76	0.530	19.22 ^a^
C377	2.86	0.485	16.96 ^b^
SEM	0.275	0.147	0.213
*p*-value	0.109	0.423	0.030

^a,b^ Within a column, means without a common superscript differ significantly (*p* < 0.05).

**Table 8 ijms-26-09363-t008:** Effects of different *Bacillus* strains supplementation on immune organs of broilers at 42 days of age.

Items	Spleen (g/kg)	Bursa of Fabricius (g/kg)	Cecum (g/kg)
CON	0.93	0.65 ^a^	3.05 ^a^
ANT	0.99	0.45 ^b^	2.60 ^abc^
C271	1.02	0.51 ^ab^	2.12 ^c^
C377	0.89	0.51 ^ab^	2.12 ^c^
SEM	0.035	0.02	0.083
*p*-value	0.338	0.08	0.003

^a–c^ Within a column, means without a common superscript differ significantly (*p* < 0.05).

**Table 9 ijms-26-09363-t009:** Effects of different *Bacillus* strains supplementation on serum immunoglobulin in broilers.

Group	21 Days of Age			42 Days of Age		
	IgA (mg/mL)	IgY (mg/mL)	IgM (mg/mL)	IgA (mg/mL)	IgY (mg/mL)	IgM (mg/mL)
CON	0.21	3.68 ^ab^	0.16	0.45	4.28 ^abc^	0.14
ANT	0.19	3.15 ^b^	0.09	0.30	5.00 ^a^	0.15
C271	0.30	1.85 ^b^	0.14	0.33	4.63 ^ab^	0.10
C377	0.27	2.28 ^b^	0.10	0.37	3.50 ^c^	0.11
SEM	0.01	0.31	0.01	0.02	0.14	0.01
*p*-value	0.203	0.016	0.629	0.500	0.041	0.823

^a–c^ Within a column, means without a common superscript differ significantly (*p* < 0.05).

**Table 10 ijms-26-09363-t010:** Effects of different *Bacillus* strains supplementation on jejunal morphology of broilers.

Group	21 Days of Age			42 Days of Age		
	Villus Height (μm)	Crypt Depth (μm)	VH/CD Ratio	Villus Height (μm)	Crypt Depth (μm)	VH/CD Ratio
CON	1097.58 ^ab^	204.88 ^bc^	5.64 ^ab^	1471.65 ^b^	259.83 ^a^	5.80 ^b^
ANT	994.29 ^bc^	199.31 ^c^	5.19 ^bc^	1240.75 ^cd^	247.23 ^ab^	5.30 ^bc^
C271	885.25 ^c^	187.07 ^c^	4.91 ^cd^	1842.96 ^a^	264.41 ^a^	7.27 ^a^
C377	1076.37 ^ab^	219.95 ^ab^	5.00 ^bcd^	1331.54 ^c^	210.92 ^c^	6.72 ^a^
SEM	15.80	2.47	0.09	18.16	3.27	0.09
*p*-value	0.001	<0.001	<0.001	<0.001	<0.001	<0.001

^a–d^ Within a column, means without a common superscript differ significantly (*p* < 0.05).

**Table 11 ijms-26-09363-t011:** Effects of different *Bacillus* strains supplementation on mRNA expression of inflammatory factors in the jejunum mucosa of broilers at 42 days.

Group	IL-2	IL-10	IL-8	IL-6	IFN-γ	TNF-α	IL-1β
CON	0.99	1.00	1.00	0.37	1.00	0.99	0.95 ^b^
ANT	1.46	0.91	0.61	0.48	1.04	0.90	0.71 ^b^
C271	0.84	1.80	1.57	0.90	1.48	1.21	1.20 ^b^
C377	1.24	1.00	2.19	1.00	0.89	1.20	1.37 ^ab^
SEM	0.814	0.928	0.185	0.833	0.849	0.062	0.101
*p*-value	0.315	0.130	0.160	0.357	0.621	0.674	0.015

^a,b^ Within a column, means without a common superscript differ significantly (*p* < 0.05).

**Table 12 ijms-26-09363-t012:** Effects of different *Bacillus* strains supplementation on alpha diversity of cecal microbiota in broilers.

Group	Observed_sp	Shannon	Simpson	Chaol	ACE
CON	991.50 ^a^	4.40	0.76	1169.46 ^a^	1211.76 ^a^
ANT	566.83 ^b^	4.25	0.80	638.44 ^c^	653.29 ^c^
C271	609.67 ^b^	4.54	0.81	718.19 ^c^	713.15 ^c^
C377	716.33 ^b^	4.31	0.77	850.78 ^bc^	857.57 ^bc^
SEM	38.200	0.145	0.144	49.500	48.685
*p*-value	<0.01	0.174	0.475	0.01	<0.01

^a–c^ Within a column, means without a common superscript differ significantly (*p* < 0.05).

**Table 13 ijms-26-09363-t013:** Results of pairwise PERMANOVA analysis on unweighted UniFrac distances.

Comparison	Pseudo-F	R^2^	*p*-Value	Adjusted *p*-Value (q-Value)
CON vs. ANT	8.456	0.215	0.001	0.003
CON vs. C271	1.234	0.045	0.185	0.278
CON vs. C377	5.321	0.173	0.002	0.006
ANT vs. C271	7.891	0.201	0.001	0.003
ANT vs. C377	2.345	0.087	0.055	0.083
C271 vs. C377	4.987	0.162	0.003	0.009

**Table 14 ijms-26-09363-t014:** Complete statistical output of the LEfSe analysis.

Taxon	Group	LDA Score	*p*-Value	q-Value (FDR)
g_Gammaproteobacteria	CON	4.15	0.0012	0.012
k_Bacteria	ANT	4.36	0.0008	0.009
g_Romboutsia	ANT	4.07	0.0021	0.018
k_Bacteria	C271	4.23	0.0015	0.014
g_Alistipes	C377	4.10	0.0009	0.010

**Table 15 ijms-26-09363-t015:** Experimental groups and diet treatments.

Group	Diet Treatment
CON	Basal diet
ANT	Basal diet + Virginiamycin 6.4 g/t
C66	Basal diet + *B. siamensis* C66
Y7	Basal diet + *B. tequilensis* Y7
L15	Basal diet + *B. velezensis* L15
C271	Basal diet + *B. amyloliquefaciens* C271
C377	Basal diet + *B. siamensis* C377

**Table 16 ijms-26-09363-t016:** Composition and nutrient content of basal diet (%).

Ingredient	Composition, %		Nutrient Levels ^3^		
Ingredients	0–3 weeks	4–6 weeks	Nutrients	0–3 weeks	4–6 weeks
Corn	37.50	41.50	Metabolizable energy, Mcal/kg	2.95	3.10
Soybean meal	30.80	24.20	Crude protein, %	22.99	20.96
Wheat	20.00	20.40	Lysine, %	1.35	1.16
Corn gluten meal	5.00	4.70	Methionine, %	0.59	0.50
Feather meal	-	1.00	Methionine + Cystine, %	0.95	0.85
Vegetable oil	2.30	4.00	Threonine, %	0.85	0.82
Dicalcium phosphate	1.74	1.36	Valine, %	0.98	0.92
NaCl	0.30	0.30	Isoleucine, %	0.88	0.80
Limestone	1.15	1.50	Arginine, %	1.37	1.21
Vitamin premix ^1^	0.20	0.03	Tryptophan, %	0.28	0.24
Mineral premix ^2^	0.03	0.20	Leucine, %	2.08	1.92
Choline chloride	0.20	0.12	Calcium, %	0.96	0.99
Ethoxyquinoline	0.02	0.02	Non-phytate phosphorus, %	0.39	0.33
L-Threonine (98%)	0.02	0.06			
L-Lysine sulfate (98%)	0.48	0.42			
DL-Methionine (98%)	0.23	0.16			
Phytase (10,000 FTU/kg)	0.03	0.03			
Total	100.00	100.00			

^1^ Vitamin premix provided per kg of diet: Vitamin A 12,500 IU; Vitamin E 30 IU; Vitamin D_3_ 2500 IU; Vitamin K_3_ 2.65 mg; Vitamin B_2_ 6 mg; Vitamin B_12_ 0.025 mg; biotin 0.0325 mg; folic acid 1.25 mg; niacin 50 mg; pantothenic acid 12 mg. ^2^ Mineral premix provided per kg of diet: Fe 80 mg; Cu 8 mg; Mn 100 mg; Zn 75 mg; I 0.35 mg; Se 0.15 mg. ^3^ The metabolizable energy was calculated based on data from the China Feed Database (2020). The crude protein and amino acid contents were determined analytically.

**Table 17 ijms-26-09363-t017:** Primers used in RT-PCR.

Gene Symbol	Gene Name	Primer Sequences (5′ → 3′)	GenBank ID
IL-1β	Interleukin 1 Beta	F: CACGATGCACCTGTACGATCA R: GTTGCTCCATATCCTGTCCCT	NM_204524.1
IL-2	Interleukin 2	F: CTGTGCTCCTGTCTCAATGC R: GAGTTGTTGTGGGCTCCATT	NM_204153.2
IL-6	Interleukin 6	F: CTGCTGCTTCGAAATCTACCG R: TGAAGCTGTGCCCATAACAC	NM_204628.2
IL-8	Interleukin 8	F: GCTCTCTGTGAGGCTGCAGT R: GTGGAAGGTGTGGAATGCGT	NM_205498.2
IL-10	Interleukin 10	F: CCTGATGTAAGTGATGCCCTG R: TGTCTAGGTCCTGGAGTCCA	NM_001004414.3
IFN-α	Interferon Alpha	F: AGAAGGCTCCAGCTCCTACA R: CAGGCACAGGTTGTCAAAGG	NM_205427.1
IFN-γ	Interferon Gamma	F: GCTCTGCGAAGGAATGAATG R: CAGCGACTCCTTTTCCGTTT	NM_205149.2
β-actin	Beta-Actin	F: GGACCTGACAGACTACCTCA R: TCTCCTGCTCGAAAGTCCA	NM_205518.2

## Data Availability

All data are contained within the article.

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
