# Peer review of "Antimicrobial and Nutritional Potentials of Bacillus Strains"

_ijms, 2025, doi:10.3390/ijms26199363_

Round 1

Reviewer 1 Report

Comments and Suggestions for Authors

The paper by Tang et al. presents the results of an extensive investigation into the beneficial properties of Bacillus strains for poultry. While the authors conducted several experiments to support their conclusions, the manuscript suffers from limited transparency in both the description of the applied methods and the presentation of the results.

The manuscript has to be significantly improved before the decision.

Comments:

1. L. 456. The Data Availability Statement in its current form does not correspond to the IJMS author guidelines.

From the author guidelines (https://www.mdpi.com/journal/ijms/instructions#sequence):
New sequence information must be deposited to the appropriate database prior to submission of the manuscript. Accession numbers provided by the database should be included in the submitted manuscript. Manuscripts will not be published until the accession number is provided.

The reported study contains high-throughput sequencing data (16S rRNA metaprofiling), but the authors do not provide any links or accession numbers to the publicly available repository with sequencing data.

2. L. 320. Please provide the ARRIVE checklist with the revised manuscript.

It is currently unclear how many samples per group were included in each analysis (required by the ARRIVE guidelines, e.g., “For each analysis, report the exact value of n in each experimental group”), which software versions were used, and how the assumptions were tested, etc.

Thereby, I cannot confirm that the report conforms to the ARRIVE guidelines, as stated by the authors.

From the author guidelines (https://www.mdpi.com/journal/ijms/instructions): 
MDPI endorses the ARRIVE guidelines (arriveguidelines.org/) for reporting experiments using live animals. Authors and reviewers must use the ARRIVE guidelines as a checklist, which can be found at https://arriveguidelines.org/sites/arrive/files/documents/Author%20Checklist%20-%20Full.pdf. The journal IJMS requires authors to submit the completed checklist at submission, and it will be made available to reviewers. Editors reserve the right to reject submissions that do not adhere to these guidelines based on ethical or animal welfare concerns, or if the procedure described does not appear to be justified by the value of the work presented.

3. Materials and Methods. Please write a number of replicates for each in vitro test (anti-C. perfringens test, acid and bile resistance tests, and enzyme production tests).

4. L. 374. Please clearly write how many samples per group were taken and analysed. The current description is not transparent and does not allow for reproducibility.
I can only suggest that samples only from one bird per replicate (five birds per group) were taken, based on Figure 1. If there were only five samples per group, then how was the application of parametric tests justified (L. 431)? How were the assumptions tested for the parametric tests?

5. L. 431. The authors do not mention the management of type I error, which raises concerns about the statistical reliability of the reported findings. The authors need to reconduct hypothesis testing with appropriate statistical adjustments and a clearer description of their analytical approach.

6. L. 433. This statement should be removed, as interpreting P-values between 0.05 and 0.10 as a “trend toward significance” lacks statistical justification and may mislead readers. The authors are encouraged to adhere to the conventional threshold (P < 0.05) and omit this phrasing. Especially given that they do not report the multiple-hypothesis testing problem management.

7. L. 359-360. Please clarify how the probiotic bacteria were added to the diets in detail. I have not found any information regarding the Bacillus sphaericus in this study.

8. L. 409-410. Please list the primers used for the PCR gene expression analyses. Were the primers validated before?

9. L. 415. Please justify the use of a kit designed for DNA extraction from the soil for the DNA extraction from the fecal samples.

10. L. 420. Maybe NovaSeq 6000? Also, please consider writing the correct manufacturer of the Illumina sequencing system. If you sequenced samples at the Novogen lab in Beijing, this needs to be acknowledged differently.

11. L. 422. Please, write the exact version of the SILVA reference database and cite the corresponding papers about this database.

12. L. 424-425. Please specify here the exact alpha and beta diversity indices used. 

13. L. 424-425. What was the rarefaction depth for the diversity analyses?

14. L. 427. Here authors mention that they conducted analysis up to the genus level, but in L. 191, they mention the analysis on the species level. V3–V4 16S rRNA HTS does not allow for reliable species-level resolution. This needs to be corrected.

15. L. 173-176. These statements about the differences in beta diversity are not supported by the results of hypothesis testing. PCoA by itself cannot be used to infer statistical significance, as it is an exploratory visualization method. I strongly recommend that the authors apply pairwise PERMANOVA (or another appropriate statistical test) to formally evaluate group differences and provide robust support for their conclusions. Do not forget to adjust the results to manage type I error.

16. L. 426-428. Differential abundance analysis.

16.1. Recent bioinformatics benchmarking studies (https://doi.org/10.1038/s41467-022-28034-z) have shown that LEfSe is not the most reliable tool for differential abundance analysis, as it is prone to high false-positive rates and sensitive to data normalization and distributional assumptions. To strengthen the validity of the findings, I recommend complementing the LEfSe results with additional methods that offer greater statistical robustness and better control of false discovery rates, such as ANCOM-BC, ALDEx2, MaAsLin2, or LinDA.

16.2. How was the data normalised for the differential abundance analysis?

16.3. Did you apply any post-hoc tests to the LEfSe results, given that four different groups were compared? Without appropriate follow-up testing, it is difficult to determine whether the reported differences are statistically robust and reliable in terms of management of false positives.

17. Tables. For each table, the authors write, "Within a row, means without a common superscript differ significantly (P < 0.05)", but it seems to be meaningless. If values are compared within a row, for example, in Table 12, then the values between Shannon and Chao1 indices are compared, which is not statistically meaningful, as these indices measure different aspects of diversity and should not be directly compared within the same row.

I guess the authors meant the differences between means within columns.

Anyway, this needs to be corrected for clarity.

18. L. 75 and elsewhere. The term "16S rDNA" is outdated. Please consider writing "16S rRNA", as you did in other parts of the text. 

19. L. 146-151. Please include microphotographs of the histological slides to support the reported findings and ensure the transparency of the research.

20. Table 12. Please provide the visualisation of alpha diversity indices among the compared groups as box plots.

21. In L. 431-432, the authors state that the "data are presented as mean ± standard error (SEM)", but Tables 6-12 do not report SEM.

22. The Abstract, Results, Discussion, and Conclusions need to be rewritten according to the new results obtained with reliable data analysis methods, which should include type I error management and other approaches acknowledged in the review.

Conclusions. The manuscript in its current form does not correspond to the IJMS level. The report is not transparent. The authors do not report false positive management. The manuscript does not conform to the ARRIVE guidelines and does not have the link to the publicly available repository with new sequencing data, which is required by the journal.

Reviewer 2 Report

Comments and Suggestions for Authors

Manuscript ijms-3852497 entitled “Screening of antimicrobial Bacillus strains and evaluation of their
Efficacy in broiler diets
”. Please notice the following:

General view: The manuscript highlighted that the newly isolated B. amyloliquefaciens C271 exhibits strong probiotic properties in vitro and enhances broiler growth and gut function in-vivo, suggesting its viability as an antibiotic growth promoter substitute. A certain degree of copyediting and proofreading, modifying the title,  adjusting the introduction, and providing some explanation in the materials and methods must be carried out before resubmission to enhance readability and understanding, and achieve the publication value.

Title: Preferred to be modified into “Antimicrobial and Nutritional Potentials of Bacillus strains” for simplicity.

Abstract: Please note the following:

  1. Clear to a certain extent.
  2. Some modifications were suggested to enhance the readability of the text.

Keywords: Modify and rearrange in alphabetical order as follows: “Bacillus; Clostridium perfringensinal Microbiota; Growth Performance; Probiotics”.

Introduction: Please note the following:

  1. Improperly arranged into four paragraphs.
  2. The introduction has to be rearranged into three paragraphs only, i.e., Introduction 2. Significance of the study, and 3. Aim of the study.
  3. Some modifications must be carried out to enhance the readability and understanding of the text.

The aim: MUST be more detailed and informative. Rewrite more appropriately.

Materials and Methods: Please note the following:

  1. How many cecal samples were collected for the in-vitro study?
  2. Illustrate with details the selective media that were used with reference to the manufacturer and source of each type.
  3. Illustrate the source and manufacturer of the LB medium.
  4. MUST specify the microclimatic conditions for raising the experimental birds, i.e., temperature, relative humidity, watering system (Type and source), feeding regimen, ventilation system, lighting program (color, intensity, and duration), cleaning program, disinfection protocol, treatment/vaccination if any.
  5. How many cecal samples were collected for the in-vivo study?
  6. MUST provide a reference for the calculations of the growth performance, organs, and slaughtering indices.
  7. Provide a reference for the software used for the statistical analysis and the statistical model used.
  8. Some modifications must be carried out to enhance the readability and understanding of the text.

Results: Please note the following:

  1. The authors did not calculate any frequencies for the isolated Bacillus strains from their different sources compared to the total isolates.
  2. Abbreviate the scientific names after their first appearance.
  3. Tables 1, 2, 3, 4, 5, and 7 did not show any statistics for a prominent comparison.
  4. Do not display the P-value in case of the non-significant results.
  5. Some modifications must be carried out to enhance the readability and understanding of the text.

Discussion: Please note the following:

  1. Moderate level of speculation and comparisons.
  2. Some modifications must be carried out to enhance the readability and understanding of the text.

Conclusion: Some modifications must be carried out to enhance the readability of the text.

Authors’ contributions: Clear and informative.

References: Sufficient, as 76.7% (43 out of 56) were published in the past five years.

Tables: Reduce the number of tables by merging the tables with similar heads.

Figures: Well-organized and presented.

Attached are the manuscript files with all the suggested modifications in TRACK AND CHANGE.

Comments on the Quality of English Language

Language: The manuscript was expressed using moderate English and grammar.

Round 2

Reviewer 1 Report

Comments and Suggestions for Authors

The authors addressed most of the comments properly; however, there remain serious issues regarding data transparency and the clarity of the data analysis outcomes.

1. In Response 1, the authors provided accession numbers to the Sanger sequencing data of the 16S rRNA gene from Bacillus bacteria used as probiotics in the study. However, they did not provide access to the raw high-throughput V3–V4 16S rRNA sequencing data, which is far more valuable for reproducibility, re-analysis, and meta-analysis purposes. Depositing this dataset in a public repository is essential to ensure transparency.

Thus, the manuscript still does not correspond to IJMS's requirements (https://www.mdpi.com/journal/ijms/instructions#sequence).

2. In the revision, the authors acknowledge that they did not apply any correction for multiple hypothesis testing in the LEfSe analysis. This omission is highly concerning, particularly given that differential abundance was assessed across four groups, where the risk of false positives is substantial.
I acknowledge that the authors mentioned this limitation in the results section; however, they did not revise the discussion accordingly. Instead, they continue to rely heavily on outcomes that are highly prone to false positives, which may mislead readers, particularly those without a strong statistical background.
If the authors are aware that applying multiple comparison adjustments would reduce the statistical significance of their microbiota composition findings, this is not critical. The study still shows improvements in production performance, which are of greater practical value for agriculture.

I strongly recommend applying multiple comparison corrections, as I cannot imagine the publication of results with such a high risk of false positives in a Q1 journal with IF = 4.9. In the context of LEfSe analysis, the authors could consider setting a more relaxed threshold for adjusted outcomes (e.g., α = 0.1 or even 0.2) and justify this choice. This approach has been adopted in multiple published studies. There is nothing wrong with such an approach, as long as it ensures statistical consistency and transparency. Importantly, I also ask the authors to report the exact p-values and corresponding q-values for all differential abundance outcomes (preferably on Figure 2), so that readers can assess the robustness of the findings directly.

Reviewer 2 Report

Comments and Suggestions for Authors

Manuscript ijms-3852497R1 entitled “Antimicrobial and Nutritional Potentials of Bacillus strains”. Please notice the following:

General view: The authors revised the manuscript professionally and highlighted that the newly isolated B. amyloliquefaciens C271 exhibits strong probiotic properties in vitro and enhances broiler growth and gut function in-vivo, suggesting its viability as an antibiotic growth promoter substitute.

Language: The manuscript was expressed using good English and grammar.

Title: Concise and clear.

Abstract: Clear and informative.

Keywords: Properly arranged and clear.

Introduction: Clear, comprehensive, and informative.

The aim: Clear.

Materials and Methods: Brief, clear, and repeatable.

Results: Novel and clear.

Discussion: Good level of speculation and comparisons, and contributes to knowledge.

Conclusion: Clear.

Authors’ contributions: Clear and informative.

References: Sufficient, as 70.7% (46 out of 65) were published in the past five years.

Tables: Well-organized and presented.

Figures: Well-organized and presented.

Round 3

Reviewer 1 Report

Comments and Suggestions for Authors

The authors addressed all the comments; however, their response regarding the raw 16S rRNA HTS data remains a concern. Therefore, I leave the decision to the Editor. I hope my comments have contributed to improving the quality of the report.